# All your loss are belong to Bayes

**Christian Walder** [1] [2]     **Richard Nock** [1] [2]

[1] CSIRO Data61, Australia.
[2] The Australian National University.

`{first.last}@data61.csiro.au`

## Abstract

Loss functions are a cornerstone of machine learning and the starting point of most algorithms. Statistics and Bayesian decision theory have contributed, via *properness*, to elicit over the past decades a wide set of admissible losses in supervised learning, to which most popular choices belong (logistic, square, Matsushita, *etc.*). Rather than making a potentially biased *ad hoc* choice of the loss, there has recently been a boost in efforts to *fit* the loss to the domain at hand while training the model itself. The key approaches fit a canonical link, a function which monotonically relates the closed unit interval to $\mathbb{R}$ and can provide a proper loss via integration.

In this paper, we rely on a broader view of proper *composite* losses and a recent construct from information geometry, *source* functions, whose fitting alleviates constraints faced by canonical links. We introduce a trick on squared Gaussian Processes to obtain a random process whose paths are compliant source functions with many desirable properties in the context of link estimation. Experimental results demonstrate substantial improvements over the state of the art.

## 1   Introduction

Supervised learning has been an influential framework of machine learning, swayed decades ago by pioneers who managed to put into equations its nontrivial uncertainties in sampling and model selection [Val84, Vap98]. One lighthearted but famous videogame-born summary[1] comes with an unexpected technical easter egg: uncertainty grasps the crux of the framework, *the loss*. All these works have indeed been predated from the early seventies by a rich literature on admissible losses for supervised learning, in statistical decision theory [Sav71], and even earlier in foundational philosophical work [dF49]. The essential criterion for admissibility is *properness*, the fact that Bayes' rule is optimal for the loss at hand. Properness is *intensional*: it does not provide a particular set of functions to choose a loss from. Over the past decades, a significant body of work has focused on eliciting the set of admissible losses (see [NM20] and references therein) ; yet in comparison with the vivid breakthroughs on models that has flourished during the past decade in machine learning, the decades-long picture of the loss resembles a still life — more often than not, it is fixed from the start, *e.g.* by assuming the popular logistic or square loss, or by assuming a restricted parametric form [Cza97, CM00, NDF00, CR02]. More recent work has aimed to provide machine learning with greater flexibility in the loss [HT92, KS09, KKSK11, NM20] — yet these works face significant technical challenges arising from (i) the joint estimation non-parametric loss function along with the remainder of the model, and (ii) the specific part of a proper loss which is learned, called a link function, which relates class probability estimation to real-valued prediction [RW10].

*In a nutshell*, the present work departs from these chiefly frequentist approaches and introduces a new and arguably Bayesian standpoint to the problem (see [GRR] for a related discussion on the nature of such a standpoint). We exploit a finer characterisation of loss functions and a trick on the Gaussian Process (GP) for efficient inference while retaining guarantees on the losses. Experiments show that our approach tends to significantly beat the state of the art [KS09, KKSK11, NM20], and records better results than baselines informed with specific losses or links, which to our knowledge is a first among approaches learning a loss or link.

More specifically, we first sidestep the impediments and constraints of fitting a link by learning a *source* function, which need only be monotonic, and which allows to reconstruct a loss via a given link — yielding a proper *composite* loss [RW10]. The entire construct exploits a fine-grained information-geometric characterisation of Bregman divergences pioneered by Zhang and Amari [Ama12, Ama13, Zha04]. This is our **first contribution**. Our **second contribution** exploits a Bayesian standpoint on the problem itself: since properness does not specify *a* loss to minimise, we do not estimate nor model *a* loss based on data — instead we perform Bayesian inference on the losses themselves, and more specifically on the source function. From the losses standpoint, our technique brings the formal advantage of a fine-tuned control of the key parameters of a loss which, in addition to properness, control consistency, robustness, calibration and rates.

We perform Bayesian inference within this class of losses by a new and simple trick addressing the requirement of priors over functions which are simultaneously non-parametric, differentiable and guaranteed monotonic. We introduce for that purpose the Integrated Squared Gaussian Process (ISGP), which is itself of independent interest as a convenient model for monotonic functions. The ISGP extends by integration (which yields monotonicity) the *squared* Gaussian Process — which has itself seen a recent burst of attention as a model for merely *non-negative* functions [ST03, MM06, LGOR15, WB17, FTS17, LH19]. The ISGP contrasts with previous approaches to learning monotonic functions which are either non-probabilistic [ABE+55, YW09, APS10, LR14, Bac18, Lim18], resort to a discretisation [HHLK19, KEC19, UKEC19], or are only approximately monotonic due to the use of only a finite set of monotonicity promoting constraints [RV10, GBCC15, SPV16, LRG+17, YLR+18, ANARL19].

▷ **Organisation.** In Section 2 we introduce properness and *source functions*. We define the ISGP model in Section 3, and provide relevant Bayesian inference procedures in Section 4. Numerical experiments are provided in Section 5 before concluding with Section 6. Technical details and an extensive suite of illustrations are provided in the supplementary appendices.

## 2 Definitions and key properties for losses

Our notations follow [NM20, RW10]. In the context of statistical decision theory as discussed more than seventy years ago [dF49] and later theorised [Sav71], the key properties of a supervised loss function start with ensuring that Bayes' rule is optimal for the loss, a property known as *properness*.

▷ **Proper (composite) losses**: let $\mathcal{Y} \doteq \{-1, 1\}$ be a set of labels. A loss for binary class probability estimation [BSS05] is a function $\ell : \mathcal{Y} \times [0, 1] \to \overline{\mathbb{R}}$ (closure of $\mathbb{R}$). Its *(conditional) Bayes risk* function is the best achievable loss when labels are drawn with a particular positive base-rate,

$$\underline{L}(\pi) \doteq \inf_u \mathbb{E}_{\mathsf{Y} \sim \mathrm{Bernoulli}(\pi)} \ell(\mathsf{Y}, u),$$

where $\Pr[\mathsf{Y} = 1] = \pi$. A loss for class probability

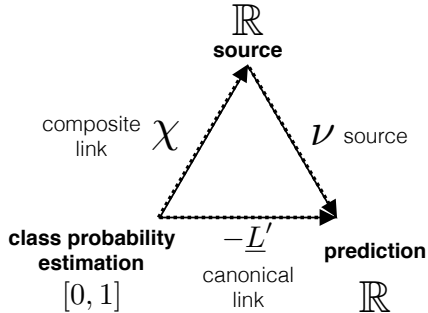

Figure 1: Relationship between source, canonical link and composite link defined by (3). Domains and names are given in the context of supervised learning.

estimation $\ell$ is ***proper*** iff Bayes prediction locally achieves the minimum everywhere: $\underline{L}(\pi) = \mathbb{E}_{\mathsf{Y}} \ell(\mathsf{Y}, \pi), \forall \pi \in [0, 1]$, and strictly proper if Bayes is the unique minimum. Fitting a classifier's prediction $h(\boldsymbol{z}) \in \mathbb{R}$ ($\boldsymbol{z}$ being an observation) into some $u \in [0, 1]$ is done via an invertible *link function* $\chi : [0, 1] \to \mathbb{R}$ connecting real valued prediction and class probability estimation. A proper loss augmented with a link, $\ell(y, \chi^{-1}(x))$ with $x \in \mathbb{R}$ [RW10] is called *proper composite*. There exists a particular link uniquely defined (up to multiplication or addition by a scalar [BSS05]) for any proper loss, the *canonical link*, as: $\chi \doteq -\underline{L}'$ [RW10, Section 6.1]: for example, the logistic loss

yields the canonical link $\chi(u) = (-\underline{L}')(u) = \log(u/(1-u))$, with inverse the well-known sigmoid $\chi^{-1}(x) = (-\underline{L}')^{-1}(x) = (1+\exp(-x))^{-1}$. A proper loss augmented with its canonical link instead of any composite link is called *proper canonical*. We remind that the Bayes risk of a proper loss is concave [RW10]. There exists a particularly useful characterisation of proper composite losses [NM20, Theorem 1]: a loss is proper composite iff

$$\ell(y, \chi^{-1}(x)) = D_{-\underline{L}}\left(y^* \| \chi^{-1}(x)\right) = D_{(-\underline{L})^\star}\left(-\underline{L}' \circ \chi^{-1}(x) \| -\underline{L}'(y^*)\right), \tag{1}$$

where $y^* \doteq (y+1)/2 \in \{0,1\}$, $D_{-\underline{L}}$ is the Bregman divergence with generator $-\underline{L}$ and $(-\underline{L})^\star$ is the Legendre conjugate of $-\underline{L}$. Let $G$ be convex differentiable. Then we have $D_G(x\|x') \doteq G(x) - G(x') - (x-x')G'(x')$ [Bre67] and $G^\star(x) \doteq \sup_{x' \in \text{dom}(G)}\{xx' - G(x')\}$ [BV04]. We assume in this paper that losses are twice differentiable for readability (differentiability assumption(s) can be alleviated [RW10, Footnote 6]) and strictly proper for the canonical link to be invertible, both properties being satisfied by popular choices (log, square, Matsushita losses, *etc.*).

▷ **Margin losses**: an important class of losses for real-valued prediction are functions of the kind $G(yh)$ where $G : \mathbb{R} \to \mathbb{R}$, $y \in \{-1, 1\}$ and $yh$ is called a margin in [RW10]. There exists a rich theory on the connections between margin losses and proper losses: If $G(x) \doteq (-\underline{L})^\star(-x)$ for a proper loss $\ell$ whose Bayes risk is symmetric around $1/2$ (equivalent to having no class-conditional misclassification costs), then minimising the margin loss is equivalent to minimising the proper canonical loss [NN08, Lemma 1]. In the more general case, given a couple $(G, \chi)$, there exists a proper loss $\ell$ such that $G(y\chi(u)) = \ell(y, u)$ ($\forall y, u$) iff [RW10, Corollary 14]:

$$\chi^{-1}(x) = \frac{G'(-x)}{G'(-x) + G'(x)}, \forall x. \tag{2}$$

When this holds, we say that the couple $(G, \chi)$ is *representable* as a proper loss. When properness is too tight a constraint, one can rely on *classification calibration* [BJM06], which relaxes the optimality condition on just predicting labels. Such a notion is particularly interesting for margin losses. We assume in this paper that margin losses are twice differentiable, a property satisfied by most popular choices (logistic, square, Matsushita, *etc.*), but notably not satisfied by the hinge loss.

▷ **Key quantities for a good loss**: there are two crucial quantities that indicate the capability of a function to be a good choice for a loss, its Lipschitz constant and weight function. For any function $G : \mathbb{R} \to \mathbb{R}$, the Lipschitz constant of $G$ is $\text{lip}_G \doteq \sup_{x,x'} |G(x) - G(x')|/|x - x'|$ and weight function $w_G(x) \doteq G''(x)$. Controlling the Lipschitz constant of a loss is important for robustness [CBG$^+$17] and mandatory for consistency [Tel13]. The weight function, on the other hand, defines properness [Sch89] but also controls optimisation rates [KM99, NN08] and defines the geometry of the loss [AN00, Section 3]. More than the desirable properness, in fitting or tuning a loss, one ideally needs to make sure that levers are easily accessible to control these two quantities.

▷ $(u, v)$-**geometric structure**: a proper loss defines a canonical link. A proper composite loss therefore makes use of *two* link functions. Apart from flexibility, there is an information geometric justification to such a more general formulation, linked to a finer characterisation of the information geometry of Bregman divergences introduced in [Ama12, Ama13, Zha04] and more recently analysed, the $(u, v)$-geometric structure [NNA16]. $u, v$ are two differentiable invertible functions and the gradient of the generator of the divergence is given by $u \circ v^{-1}$. This way, the two dual forms of a Bregman divergence (in (1)) can be represented using the same coordinate system known as the *source*, which specifies the dual coordinate system and Riemannian metric induced by the divergence. We consider the $(\nu, \chi)$-geometric structure of the divergence $D_{-\underline{L}}$, which therefore gives for $\nu$:

$$\nu = -\underline{L}' \circ \chi^{-1}, \tag{3}$$

and yields a local composite estimate of Bayes' rule for a corresponding $x \in \mathbb{R}$ as in [NM20, Eq. 4]:

$$\hat{p}(y = 1|x; \nu, \underline{L}) \doteq \chi^{-1}(x) = (-\underline{L}')^{-1} \circ \nu(x). \tag{4}$$

Figure 1 depicts $\nu$, which we call "source" as well. The properties of $\nu$ in (3) are summarised below.

**Definition 1.** *A **source** (function) $\nu$ is a differentiable, strictly increasing function.*

Any loss completed with a source gives a proper composite loss. There are two expressions of interest for a loss involving source $\nu$, which follow from (1) and margin loss $G(.)$, with $y \in \{0, 1\}$ and $x \in \mathbb{R}, u \in [0, 1]$ (with the correspondence $x \doteq \chi(u)$ and $y^* \doteq (y+1)/2$):

$$\ell_\nu(y, u) \doteq D_{(-\underline{L})^\star}\left(\nu(x)\| -\underline{L}'(y^*)\right) \tag{5}$$

$$G_\nu(y\chi(u)) \doteq G(y \cdot (\nu^{-1} \circ (-\underline{L}'))(u)). \tag{6}$$

When it is clear from the context, we shall remove the index "$\nu$" for readability. Instead of learning the canonical link of a loss, which poses significant challenges due to its closed domain [KKSK11, NM20], we choose to infer the source given a loss. As we now see, this can be done efficiently using Gaussian Process inference and with strong guarantees on the losses involved.

## 3 The Integrated Squared Gaussian Process

▷ **Model Definition.** An Integrated Squared Gaussian Processes (ISGP) is a non-parametric random process whose sample paths turn out to be source functions.

**Definition 2.** *An ISGP is defined by the following model:*

$$\nu_0 \sim \mathcal{N}(\mu, \gamma^{-1}); \qquad f \sim \mathrm{GP}(k(\cdot, \cdot)) \quad \Leftrightarrow \quad \begin{cases} f(\cdot) = \boldsymbol{w}^\top \boldsymbol{\varphi}(\cdot) \\ \boldsymbol{w} \sim \mathcal{N}(0, \Lambda) \end{cases}$$

*and $\nu(x) \doteq \nu_0 + \int_0^x f^2(z)\,\mathrm{d}z$.*

Here, $\boldsymbol{\varphi}(x) = (\varphi_1(x), \varphi_2(x), \ldots, \varphi_M(x))^\top$ represents $M$ basis functions, $\Lambda \in \mathbb{R}^{M \times M}$ is a diagonal matrix with elements $\lambda_m > 0$, and $\mathrm{GP}(k(\cdot, \cdot))$ is the zero-mean Gaussian process with kernel $k : \mathbb{R} \times \mathbb{R} \to \mathbb{R}$. Hence the above equivalence (denoted $\Leftrightarrow$) implies the Mercer expansion

$$k(x, z) = \boldsymbol{\varphi}(x)^\top \Lambda \boldsymbol{\varphi}(z). \tag{7}$$

Note that while for finite $M$ this construction implies a *parametric* model, parameteric approximations are common in GP literature [RW05]. The ISGP is Gaussian in the parameters $\boldsymbol{w}$ and $\nu_0$, and admits a simple form for $\nu$:

$$\nu(x) = \nu_0 + \int_0^x f^2(z)\,\mathrm{d}z = \nu_0 + \int_0^x \left(\boldsymbol{w}^\top \boldsymbol{\varphi}(z)\right)^2 \mathrm{d}z = \nu_0 + \boldsymbol{w}^\top \boldsymbol{\psi}(x)\boldsymbol{w}, \tag{8}$$

with the positive semi-definite matrix

$$\boldsymbol{\psi}(x) \doteq \int_0^x \boldsymbol{\varphi}(z)\boldsymbol{\varphi}(z)^\top \,\mathrm{d}z \in \mathbb{R}^{M \times M}. \tag{9}$$

It is our choice of *squaring* of $f$ — previously exploited to model *non-negativity* [ST03, MM06, LGOR15, WB17, FTS17, LH19] — which leads to this simple form for the *monotonic $\nu$*.
▷ **Sanity checks of the ISGP prior for loss inference.** We now formally analyse the goodness of fit of an ISGP in the context of loss inference. The following Lemma is immediate and links the ISGP to inference on functions expressed as Bregman divergences, and therefore to proper losses (Section 2).

**Lemma 1.** *$\nu(x)$ as in Definition 2 is a source function with probability one.*

*Remark* 3.1. Bregman divergences are also the analytical form of losses in other areas of machine learning (*e.g.* unsupervised learning [BGW05]), so ISGPs may *via* Lemma 1 be of broader interest.

The following Theorem safeguards inference on loss functions with an ISGP.

**Theorem 2.** *For any convex $G$ and any ISGP, the following holds with probability one: (i) if $G$ is decreasing, there exists a canonical link $-\underline{L}'$ such that the couple $(G, \chi)$ where $\chi$ is obtained from (3) is representable as a proper loss; (ii) if $G$ is classification calibrated, the margin loss $G_\nu(x) \doteq G \circ \nu^{-1}(x)$ is classification calibrated.*

(Proof in Appendix C.) To summarise, the whole support of an ISGP is fully representable as proper losses, but the proof of point (i) is not necessarily constructive as it involves a technical invertibility condition. On the other hand, part (ii) relaxes properness for classification calibration but the invertibility condition is weakened to that of $\nu$ for the margin loss involved.

Theorem 2 (and the title of the present work) begs for a converse on the condition(s) on the ISGP to ensure that *any* proper composite loss can be represented in its support. Fortunately, a sufficient condition is already known [MXZ06].

**Theorem 3.** *Suppose kernel $k$ in Definition 2 is universal. Then for any proper loss $\ell$ and any composite link $\chi$, there exists $\nu$ as in Definition 2 such that the proper composite loss $\ell(y, \chi^{-1}(x))$ can be represented as in (5): $\ell(y, \chi^{-1}(x)) = \ell_\nu(y, u), \forall u \in [0, 1], y \in \{-1, 1\}$ and $x \doteq \chi(u)$.*

To avoid a paper laden paper with notation, we refer to [MXZ06] for the exact definition of universality which, informally, postulates that the span of the kernel can approximate any function to arbitrary precision. Interestingly, a substantial number of kernels are universal, including some of particular interest in our setting (see below). We now investigate a desirable property of the first-order moment of a loss computed from an ISGP. We say that the ISGP is *unbiased* iff $\mathbb{E}_\nu\left[\nu(x)\right] = x, \forall x$.

**Theorem 4.** *Letting $\{(x, y)\} \subset \mathbb{R} \times \mathcal{Y}$ be a sample of labelled training values. For any unbiased ISGP and any proper canonical loss $\ell$, the proper composite loss $\ell$ formed by $\ell$ and composite link $\chi \doteq \nu^{-1} \circ -\underline{L}'$ (3) satisfies: $\mathbb{E}_{(x,y)}\left[\ell(y, (-\underline{L}')^{-1}(x))\right] \leq \mathbb{E}_{(x,y),\nu}\left[\ell_\nu(y, \chi^{-1}(x))\right]$.*

(Proof in Appendix C.) In the context of Bayesian inference this means that the prior uncertainty in $\nu$ induces a prior on losses whose expected loss upper bounds that of the canonical link. We exploit this property to *initialise* our Bayesian inference scheme with fixed canonical link (see Section 4.2). Of course, this property follows from Theorem 4 *if* the ISGP is unbiased, which we now show is trivial to guarantee. We say that a kernel $k$ is translation invariant iff $k(x, x') = k(0, x - x'), \forall x, x'$.

**Theorem 5.** *For any translation invariant $k$, the ISGP $\nu$ satisfies $\mathbb{E}_\nu\left[\nu(x)\right] = \mu + k(0, 0) \cdot x$.*

*Proof.* By linearity of expectation this mean function equals $\mathbb{E}[\nu_0] = \mu$ plus the simple term

$$\mathbb{E}_{f \sim \text{GP}(k(\cdot, \cdot))}\left[\int_0^x f^2(z)\,\mathrm{d}z\right] = \int_0^x \mathbb{E}\left[f^2(z)\right]\mathrm{d}z = \int_0^x k(z, z)\,\mathrm{d}z = x \cdot k(0, 0), \quad (10)$$

yielding the statement of the Theorem. $\qquad\square$

Hence, if $\mu = 0$ and $k(0, 0) = 1$ in Definition 2, the ISGP is unbiased. Interestingly, some translation invariant kernels are universal [MXZ06, Section 4]. We now dig into how an ISGP allows to control the properties of supervised losses that govern robustness and statistical consistency, among others [CBG$^+$17, Tel13]. Denote $\|\varphi\|^2_{\max} = \sup_x \|\varphi(x)\|^2_2$, which is simple to upper-bound for the $\varphi$ we will adopt later (see (14)).

**Lemma 6.** *For ISGP $\nu$ the Lipschitz constant of $\ell_\nu$ in (5) satisfies $\text{lip}_{\ell_\nu} \leq \|\varphi\|^2_{\max} \cdot \|w\|^2_2 \cdot \text{lip}_\ell$.*

*Proof.* By Cauchy-Schwartz $\nu'(x) = f^2(x) = (w^\top \varphi(x))^2 \leq \sup_x(w^\top \varphi(x))^2 \leq \sup_x \|\varphi(x)\|^2_2 \cdot \|w\|^2_2 \leq \|\varphi\|^2_{\max} \cdot \|w\|^2_2$. The Lemma can then follows from the chain rule on $\ell_\nu$ in (5). $\qquad\square$

Similarly, we have $\chi = \nu^{-1} \circ (-\underline{L}')$ from (3) and so $\chi'(u) = w_\ell(u) \cdot (w^\top(\varphi\varphi^\top)(\chi(u))w)^{-1}$, yielding this time a Lipschitz constant for $G_\nu$ of (6) which is proportional to the weight of $\ell$ and inversely proportional to $f^2$. This shows that the prior's uncertainty still allows for a probabilistic handle on the Lipschitz constant, $\Lambda$ — the eigenspectrum in the Mercer expansion.

## 4 Inference with Integrated Squared Gaussian Processes

In Section 4.1 we give an approximate inference method for the ISGP with simple i.i.d. likelihood functions — this is analogous to the basic GP regression and classification in *e.g.* [RW05]. The subsequent Section 4.2 builds on this to provide an inference method for proper losses for the generalised linear model — this is a Bayesian analog of the loss fitting methods in [KKSK11, NM20].

### 4.1 Basic Laplace-approximate ISGP inference for a fixed likelihood function

Efficient variational inference for the ISGP is an open problem due to the intractable integrals this entails. It is straight-forward however to construct a Laplace approximation to the posterior in the parameters $\Gamma = \{w, \nu_0\}$, provided that *e.g.* the likelihood function for the data $\mathcal{D} = \{x_n, y_n\}_{n=1}^N$ factorises in the usual way as $\log p(\mathcal{D}|\nu, \Theta) = \sum_{n=1}^N \log p(y_n|\nu(x_n), \Theta)$. Here $\Theta$ represents the hyper-parameters — which are $\gamma, \mu$, any parameters of the kernel $k(\cdot, \cdot)$, and any parameters of the likelihood function. We use automatic differentiation and numerical optimisation to obtain the posterior mode $\widehat{\Gamma} = \text{argmax}_\Gamma \log p(\mathcal{D}, \Gamma)$. We then use automatic differentiation once more to compute the Hessian

$$H = \left.\frac{\partial^2}{\partial\Gamma\partial\Gamma^\top}\right|_{\Gamma=\widehat{\Gamma}} - \log p(\mathcal{D}, \Gamma). \quad (11)$$

Letting $\widehat{\Sigma}_\Gamma = H^{-1}$, the approximate posterior is then $p(\Gamma|\mathcal{D}) \approx \mathcal{N}(\Gamma|\widehat{\Gamma}, \widehat{\Sigma}_\Gamma) \doteq q(\Gamma|\mathcal{D})$.

▷ **Predictive distribution:** posterior samples of the monotonic function $\nu$ are obtained from samples $\Gamma^{(s)} = \{\boldsymbol{w}^{(s)}, \nu_0^{(s)}\}$ from $q$ via the deterministic relation (8). For the predictive mean function, we exploit the property that the predictive distribution is the sum of the Gaussian $\nu_0$ plus the well studied quadratic function of a multivariate normal. If we let $\widehat{\nu_0}$ be the element of $\widehat{\Gamma}$ corresponding to $\nu_0$, and let $\widehat{\Sigma}_{\boldsymbol{w}}$ be the sub-matrix of $\widehat{\Sigma}_\Gamma$ corresponding to $\boldsymbol{w}$, *etc.*, then the following holds.

**Lemma 7.** *Using notations just defined, the mean function may be written as*

$$\mathbb{E}_{q(\Gamma|\mathcal{D})}[\nu(x)] = \widehat{\nu_0} + \mathrm{tr}(\boldsymbol{\psi}(x)\widehat{\Sigma}_{\boldsymbol{w}}) + \widehat{\boldsymbol{w}}^\top \widehat{\Sigma}_{\boldsymbol{w}} \widehat{\boldsymbol{w}}.$$

*Proof.* We have $\mathbb{E}_{q(\Gamma|\mathcal{D})}[\nu(x)] = \widehat{\nu_0} + \mathbb{E}_{\mathcal{N}(\boldsymbol{w}|\widehat{\boldsymbol{w}}, \widehat{\Sigma}_{\boldsymbol{w}})}[\boldsymbol{w}^\top \boldsymbol{\psi}(x)\boldsymbol{w}]$ and, for $\boldsymbol{w} \sim \mathcal{N}(\boldsymbol{w}|\widehat{\boldsymbol{w}}, \widehat{\Sigma}_{\boldsymbol{w}})$,

$$\mathbb{E}[\boldsymbol{w}^\top \boldsymbol{\psi}(x)\boldsymbol{w}] = \mathbb{E}[\mathrm{tr}(\boldsymbol{\psi}(x)\boldsymbol{w}\boldsymbol{w}^\top)] = \mathrm{tr}(\boldsymbol{\psi}(x)\mathbb{E}[\boldsymbol{w}\boldsymbol{w}^\top]) = \mathrm{tr}(\boldsymbol{\psi}(x)(\widehat{\Sigma}_{\boldsymbol{w}} + \widehat{\boldsymbol{w}}\widehat{\boldsymbol{w}}^\top)),$$

which leads to the required result. $\qquad\square$

This mean function differs from that of a simple GP, which is linear w.r.t. $\widehat{\boldsymbol{w}}$. Closed form expressions for the higher moments of $\boldsymbol{w}^\top \boldsymbol{\psi}(x)\boldsymbol{w}$ are provided in [MP92, §3.2b: *Moments of a Quadratic Form*].

We can easily replicate the use of Jensen's inequality in the proof of Theorem 4 to obtain the following guarantee for the posterior mean function. We recall that $\underline{L}$ is the Bayes risk of a proper loss $\ell$ and $\chi^{-1} \doteq (-\underline{L}')^{-1} \circ \nu$ is the construct of the (inverse) composite link using source $\nu$ (of (3)).

**Theorem 8.** *For any sample of labelled data points $\{(x, y)\} \subset \mathbb{R} \times \mathcal{Y}$, any proper loss $\ell$ and source $\nu$ sampled from approximate posterior $q(\Gamma|\mathcal{D})$, we have:*

$$\mathbb{E}_{(x,y),\nu}[\ell_\nu(y, \chi^{-1}(x))] \le \mathbb{E}_{(x,y)}\left[D_{(-\underline{L})^\star}\left(\widehat{\nu_0} + \mathrm{tr}(\boldsymbol{\psi}(x)\widehat{\Sigma}_{\boldsymbol{w}}) + \widehat{\boldsymbol{w}}^\top \widehat{\Sigma}_{\boldsymbol{w}} \widehat{\boldsymbol{w}}\| - \underline{L}'(y^*)\right)\right].$$

Note that the l.h.s. above takes an expectation with respect to the approximate posterior of $\nu$ while the r.h.s. involves the parameters of the approximate posterior mean function $\widehat{\nu}$. This clarifies the importance of Bayesian modelling of $\nu$, as the resulting expected loss is merely upper bounded by that for the fixed mean function. Alternatively the result suggests a cheap approximation to marginalising (4) w.r.t. $\nu$ for prediction, namely putting the mean function for $\nu$ into that equation.

▷ **Marginal likelihood approximation and optimisation:** we follow [SF06] and use automatic differentiation to achieve the same result as — while avoiding the manual gradient calculations of — the usual approach in the GP literature [RW05, §3.4]. See Section D for further details on the marginal likelihood, computational complexity, and likelihood functions for the univariate case.

## 4.2 Using the basic ISGP inference to fit Bayes estimates via Expectation Maximisation

At this stage, inferring the loss in a machine learning model boils down to simply placing an ISGP prior on the source function and doing Bayesian inference. In accordance with part i) of Theorem 2, this induces a posterior over sources which in turn implies a distribution over proper composite losses. We now present an effective inference procedure for the generalised linear model, which has been the standard model for closely related isotonic regression approaches to the problem [KS09, KKSK11, NM20]. We first choose a proper loss' link $-\underline{L}'$ to get the proper composite estimate (4). Here we consider the log loss and therefore the inverse sigmoid for $-\underline{L}'$, although we emphasise that our method works with any proper loss in which the source is embedded.

▷ **Model:** to summarise, given a dataset $\mathcal{D} = \{\boldsymbol{z}_n, y_n\}_{n=1,\dots,N} \subset \mathbb{R}^D \times \mathcal{Y}$, we model $\hat{p}(x) \doteq (-\underline{L}')^{-1} \circ \nu(x)$ from (4) with $(-\underline{L}')^{-1}(x) = (1 + \exp(-x))^{-1}$, to obtain the classification model

$$y_n | \boldsymbol{z}_n \sim \mathrm{Bernoulli}(\hat{p}(x_n)) \quad ; \quad x_n = \boldsymbol{\beta}^\top \boldsymbol{z}_n, \tag{12}$$

where $\nu \sim \mathcal{F}$ and $\mathcal{F}$ is a prior over functions such as the GP or ISGP. We leave the prior on the linear model weight vector $\boldsymbol{\beta} \in \mathbb{R}^D$ unspecified as we perform maximum (marginal) likelihood on that parameter. This model generalises logistic regression, which we recover for $\nu(x) = x$.

▷ **Inference:** we use Expectation Maximisation (EM) to perform maximum likelihood in $\boldsymbol{\beta}$, marginal of $\nu$. In accordance with Theorem 4 we initialise $\boldsymbol{\beta}^{(\mathrm{old})}$ using traditional fixed link $(-\underline{L}')^{-1}$. In the *E-step*, $x_n = \boldsymbol{\beta}^{(\mathrm{old})\top} \boldsymbol{z}_n$ are fixed and we use the Laplace approximate inference scheme of Section 4.1

to compute the posterior in $\nu$ (or equivalently $\Gamma = \{\boldsymbol{w}, \nu\}$), taking the likelihood function to be that of $y_n|x_n$ from (12), above. Let that approximate posterior — previously denoted by $q(\Gamma|\mathcal{D})$ — be denoted here by $q(\nu|\boldsymbol{\beta}^{(\text{old})})$. The *M-step* then updates $\boldsymbol{\beta}$ by (letting $\boldsymbol{y} = \{y_1, y_2, \ldots, y_N\}$ and $\boldsymbol{x} = \{x_1, x_2, \ldots, x_N\}$),

$$\boldsymbol{\beta}^{(\text{new})} = \underset{\boldsymbol{\beta}}{\arg\max}\, Q(\boldsymbol{\beta}|\boldsymbol{\beta}^{(\text{old})}) \quad ; \quad Q(\boldsymbol{\beta}|\boldsymbol{\beta}^{(\text{old})}) = \mathbb{E}_{q(\nu|\boldsymbol{\beta}^{(\text{old})})}\left[\log p(\boldsymbol{y}|\boldsymbol{x}, \nu)\right].$$

Although the expectation for $Q$ is analytically intractable, the univariate setting suggests that Monte Carlo will be effective. We draw $S$ samples $\nu^{(s)}$, $s = 1, \ldots, S$ from $q(\nu|\boldsymbol{\beta}^{(\text{old})})$, and approximate

$$\mathbb{E}_{q(\nu|\boldsymbol{\beta}^{(\text{old})})}\left[\log p(\boldsymbol{y}|\boldsymbol{x}, \nu)\right] \approx \frac{1}{S}\sum_{s=1}^{S}\sum_{n=1}^{N}\log \text{Bernoulli}(y_n|(-\underline{L}')^{-1} \circ \nu^{(s)}(\boldsymbol{\beta}^\top \boldsymbol{z}_n)),$$

where $\nu^{(s)}$ is given by (8). The above expression may be automatically differentiated and optimised using standard numerical tools. To achieve this efficiently under the ISGP prior for $\mathcal{F}$, we implement a custom derivative function based on the relation

$$\nu'(x) = f^2(x) = \left(\boldsymbol{w}^\top \boldsymbol{\varphi}(x)\right)^2, \tag{13}$$

rather than requiring the automatic differentiation library to differentiate through $\psi$ in (8).

Given our approximate posterior, probabilistic predictions are obtained by marginalising out $\nu$ in a manner analogous to a GP classifier [RW05].

▷ **Computational Complexity** Our $\mathcal{O}(M^2N)$ per EM iteration algorithm admits control of $M$, a meaningful knob to control complexity. This compares favourably with the state of the art [NM20, § 5], the *fastest* option of which costs $\mathcal{O}(N\log N)$ per iteration at the expense of a rather involved data structure, without which their procedure costs $\mathcal{O}(N^2)$.

## 4.3 The trigonometric kernel

We introduce a novel kernel designed to conveniently lend itself to the ISGP. See Appendix F for the requirements this entails, and a discussion of a tempting yet less convenient alternative (the standard Nyström approximation [RW05]). Our main insight is that we need not fix the kernel and derive the corresponding Mercer expansion — even if such an expansion is available, the integrals for $\psi(x)$ are generally intractable. Instead we recall that $k(x, z) = \boldsymbol{\varphi}(x)^\top \Lambda \boldsymbol{\varphi}(z) = \sum_{m=1}^{M} \lambda_m\, \varphi_m(x)\varphi_m(z)$, and we let the $(\lambda_m, \varphi_m(x))$ pairs (of which there are an even number, $M$) be given by the following union of two sets of $M/2$ pairs,

$$\left\{\left(b/a^m,\, \cos\left(\pi m c x\right)\right)\right\}_{m=1}^{M/2} \bigcup \left\{\left(b/a^m,\, \sin\left(\pi m c x\right)\right)\right\}_{m=1}^{M/2}, \tag{14}$$

on the domain $x \in [-1/c, +1/c]$, where $a > 1$ and $b > 0$ are input and output length-scale hyper-parameters and $M \approx 100$ is chosen large enough to provide sufficient flexibility. This is related to *i)* the construction in [WB17] (but has the advantage of admitting closed form expressions for $k$), and *ii)* the random features approaches of *e.g.* [RR08, LGQCRFV10, CBMF17] (which differ by approximating a fixed covariance with a trigonometric basis). It is easy to show that the kernel is translation invariant and that the prior variance — which is especially relevant here due to Theorem 5 — is given by

$$k(x, x) = k(0, 0) = b\left(1 - a^{-M/2}\right)/(a - 1). \tag{15}$$

We give expressions for $k(x, z)$ and $\psi(x)$ in Appendix E, and an illustration in Figure 4.

## 5 Experiments

We provide illustrative examples and quantitative comparisons of the ISGP prior in univariate regression/classification, and inferring proper losses. The code for the real world problems of learning the loss is available online.[2] We fixed $M = 64$ in (14). Classification problems used $(-\underline{L}')^{-1} = \sigma$

Table 1: Mean test set negative log likelihoods for various isotonic regression methods. See text for details.

| Train | Input | GP | ISGP | PAVA | Linear |
|-------|-------|------|------|------|--------|
| Large | accel. | 2080.9 | 2083.4 | **2055.0** | 2105.9 |
|       | displ. | 788.1 | 780.6 | **760.7** | 907.2 |
|       | power | 763.4 | 763.9 | **755.5** | 1002.2 |
|       | weight | **735.5** | 750.7 | 769.9 | 788.4 |
| Small | accel. | **2173.3** | 2216.8 | 2212.9 | 2199.6 |
|       | displ. | 871.0 | **849.4** | 871.4 | 950.9 |
|       | power | 827.3 | **811.8** | 826.4 | 1033.5 |
|       | weight | 777.2 | **770.7** | 835.8 | 815.8 |

Table 2: Test AUC for generalised linear models with various link methods (ordering in decreasing average). See text for details.

|  | mnist | fmnist |
|---|-------|--------|
| ISGP-Linkgistic | 99.9 % | 99.2 % |
| GP-Linkgistic | 99.9 % | 99.1 % |
| Logistic regression | 99.9 % | 98.5% |
| GLMTron | 99.6% | 98.1% |
| BREGMANTRON | 99.7% | 97.9% |
| BREGMANTRON$_{label}$ | 99.6% | 97.7% |
| BREGMANTRON$_{approx}$ | 99.3% | 94.6% |
| SLISOTRON | 94.6% | 90.7% |

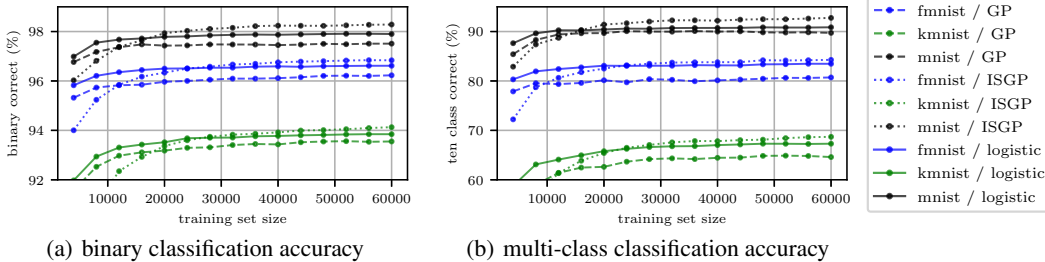

(a) binary classification accuracy      (b) multi-class classification accuracy

Figure 2: Test performance *v.s.* training set size for the *MNIST*, *Kuzushiji-MNIST* and *Fashion-MNIST* datasets. We compare logistic regression (logistic), GP-LINKGISTIC (GP) and ISGP-LINKGISTIC (ISGP). On the left we show the mean *one vs the rest* classification accuracy, and on the right we show the ten class classification accuracy obtained by combining the *one vs the rest* models.

with $\sigma(x) = 1/(1 + \exp(-x))$. Optimisation was performed with L-BFGS [BLNZ95]. Gradients were computed with automatic differentiation (with the notable exception of (13)) using PyTorch [PGM$^+$19]. The experiments took roughly two CPU months.

▷ **Univariate Regression Toy.** As depicted in Figure 3, we combined the Gaussian likelihood function (17) with both the GP and ISGP function priors for $\nu$. We investigate this setup further in the supplementary Figure 5 and Figure 6, which also depict the latent $f$ and $f^2$, which are related by $\nu = f^2$. Finally, Figure 7 visualises the marginal likelihood surface near the optimal point found by the method of Section D. See the captions for details.

▷ **Univariate Classification Toy.** We illustrate the mechanics of the classification model with ISGP prior and the sigmoid-Bernoulli likelihood function of (19). We constructed a univariate binary classification problem by composing the sigmoid with a piece-wise linear function, which allows us to compare both the inferred $\nu$ and the inferred (inverse) canonical link $\chi^{-1} = (-\underline{L}')^{-1} \circ \nu$ to their respective ground truths — see Figure 8 and the caption therein for more details.

▷ **Real-world Univariate Regression.** We compared our ISGP based regression model to *a)* GP regression with the same kernel of Section 4.3, *b)* the widely used pool adjacent violators algorithm (PAVA) [Kru64] for isotonic regression, and *c)* linear least squares. For the ISGP and GP models we used maximum marginal likelihood hyper parameters. The task was to regress each of the four real-valued features of the `auto-mpg` dataset [DG17] onto the target (car efficiency in miles per gallon). We partitioned the dataset into five splits. For the `Small` problems, we trained on one split and tested on the remaining four ; for the `Large` problems we did the converse. We repeated this all $C_1^5 = C_4^5 = 5$ ways and averaged the results to obtain Table 1. As expected the Bayesian GP and ISGP models — which have the advantage of making stronger regularity assumptions — perform relatively better with less data (the `Small` case). Our ISGP is in turn superior to the GP in that case, in line with the appropriateness of the monotonicity assumption which it captures.

▷ **Real-world Problems of Learning the Loss.** We bench-marked our loss inference algorithm of Section 4.2 on binary classification problems. Note that multi-class problems may be handled via *e.g.* a 1-vs-all or 1-vs-1 multiple coding while more direct handling may be a topic of future work. We used both the GP and ISGP function priors, and we refer to the resulting algorithms as GP-LINKGISTIC and ISGP-LINKGISTIC, respectively — only the latter of which gives rise to guaranteed

*proper* losses. To ease the computational burden, we fixed the hyper-parameters throughout. We chose set $\mu = 0$ and $\gamma = 0.01$. We set the length-scale parameter $a = 1.2$ and chose $b$ such that $k(0,0) = 1$ (using (15)) — by (10) this makes $\nu(x) = x$ the most likely source *a priori*, to roughly bias towards traditional logistic regression. For the GP we tuned $b$ for good performance on the test set (for the full 60K training examples), to give the GP an unfair advantage over the ISGP, although this advantage is small as GP-LINKGISTIC is rather insensitive to the choice of $b$.

Table 2 compares the SLISOTRON and BREGMANTRON algorithms from [KKSK11] and [NM20], respectively, along with the other baselines from the latter work. Further details on the experimental setup are provided in [NM20]. In contrast with the SLISOTRON and BREGMANTRON algorithms, our models successfully match or outperform the logistic regression baseline. Moreover, the monotonic ISGP-LINKGISTIC slightly outperforms GP-LINKGISTIC, and as far as we know records the first result beating logistic regression on this problem, by a reasonable margin on `fmnist` [NM20].

We further bench-marked ISGP-LINKGISTIC against GP-LINKGISTIC and logistic regression (as the latter was the strongest practical algorithm in the experiments of [NM20]) on a broader set of tasks, namely the three MNIST-like datasets of [LC10, XRV17, CBIK$^+$18]. We found that ISGP-LINKGISTIC dominates on all three datasets, as the training set size increases — see Figure 2 and the caption therein for more details. Figure 9 depicts an example of the learned (inverse) link functions.

## 6   Conclusion

We have introduced a Bayesian approach to inferring a posterior distribution over loss functions for supervised learning that complies with the Bayesian notion of properness. Our contribution thereby advances the seminal work of [KKSK11] and the more recent [NM20] in terms of modelling flexibility (which we inherit from the Bayesian approach) and — as a direct consequence — practical effectiveness as evidenced by our state of the art performance. Our model is both highly general, and yet capable of out-performing even the most classic baseline, the logistic loss for binary classification. As such, this represents an interesting step toward more flexible modelling in a wider machine learning context, which typically works with a loss function which is prescribed *a priori*.
Since the tricks we use essentially rely on the loss being expressible as a Bregman divergence and since Bregman divergences are also a principled distortion measure for *un*supervised learning — such as in the context of the popular $k$-means and EM algorithms — an interesting avenue for future work is to investigate the potential of our approach for unsupervised learning.

## Broader Impact

From an ethical standpoint, it is likely that any advancements in fundamental machine learning methodologies will eventually give rise to both positive and negative outcomes. The present work is sufficiently general and application independent, however, as to pose relatively little cause for immediate concern.

Nonetheless, we note one example of an optimistic take on the potential impact of the present work. The example is from the field of *quantitative criminology*, wherein it is advocated that further study of asymmetric loss functions may provide lawmakers with "procedures far more sensitive to the real consequences of forecasting errors" in the context of law enforcement [Ber11]. Such a study implies that symmetric links — like the ones derived from most common losses: logistic, square, Matsushita, *etc.* — are not a good fit for sensitive fields. Since inference on proper losses can produce non-symmetric links, the present contribution may be useful for such application fields.

## Acknowledgments and Disclosure of Funding

We thank the anonymous reviewers for their insightful feedback, and we thank *The Boeing Company* for partly funding this work.

## Footnotes

[1]See `https://tinyurl.com/y5jnurau` or search for "all your Bayes are belong to us, Vapnik quote".

[2] https://gitlab.com/cwalder/linkgistic

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
