[Supplementary Material]

The following supplementary appendices accompany *All your loss are belong to Bayes*.

## A    Symbols

| | |
|---|---|
| $\mathcal{y}$ | $\doteq \{-1, 1\}$, real-valued labels |
| $\mathcal{y}^*$ | $\doteq \{0, 1\}$, Boolean labels |
| $\ell, \ell_\nu$ | (proper) loss, loss depending on source $\nu$ |
| $\underline{L}$ | (conditional) Bayes risk |
| $-\underline{L}', \chi$ | canonical link, composite link |
| $\sigma$ | sigmoid function |
| $\nu$ | source |
| $G, G_\nu$ | margin loss, margin loss depending on source $\nu$ |
| $D_G$ | Bregman divergence with (convex) generator $G$ |
| $G^\star$ | Legendre conjugate ($G$ differentiable) |
| $\nu_0; \mu, \gamma$ | ISGP constant; prior mean, prior precision (of constant) |
| $f; \boldsymbol{w}$ | GP function and Mercer weights |
| $k(\cdot, \cdot); \boldsymbol{\varphi}, \Lambda$ | GP kernel, kernel basis, kernel eigenvalues |
| $M$ | size of the Mercer expansion |
| $\boldsymbol{\psi}$ | positive-definite integral of the kernel feature's outer product |
| $\Theta$ | collected ISGP *hyper*-parameters *i.e.* $\gamma, \mu, k(\cdot, \cdot)$ + likelihood params |
| $\Gamma \doteq \{\boldsymbol{w}, \nu_0\}$ | collected ISGP parameters |
| $q$ | approximate posterior distribution |
| $\widehat{\Gamma}, \widehat{\Sigma}_\Gamma$ | approximate posterior mean, covariance (here, of $\Gamma$) |
| $\alpha$ | univariate Gaussian likelihood precision |
| $\boldsymbol{\beta}$ | generalised linear model weight vector |

## B    On the Origin Problem and an Alternative Solution

Here we discuss the role of the random intercept $\nu_0$ in Definition 2 and offer an alternative formulation.

While the integration and squaring transformations guarantee monotonicity, the question remains where to integrate from. To see this, omit the random intercept $\nu_0$, denote by $r$ the l.h.s. lower limit of integration, and define $\nu^{(r)}(x) = \int_r^x f^2(z)\, \mathrm{d}z$. For GP distributed $f$, this construction induces an artefact in the distribution for $\nu^{(r)}$, which is roughly speaking the fact that $\nu^{(r)}(r) = 0$ with probability one.

In the main text, we alleviate this issue by introducing the random intercept $\nu_0$. We offer here an alternative solution, the idea of which is to shift the starting point of integration outside the domain of interest. Assume without loss of generality that the domain of interest is the positive real line. Concretely, the alternative formulation would model $\hat{\nu}^{(r)}(x)$ as, for $r \leq 0$,

$$\hat{\nu}^{(r)}(x) = \int_r^x f^2(z)\, \mathrm{d}z - \mu(r)$$

$$f \sim \mathrm{GP}(k(\cdot, \cdot)) \Leftrightarrow \begin{cases} f(\cdot) = \boldsymbol{w}^\top \boldsymbol{\varphi}(\cdot) \\ \boldsymbol{w} \sim \mathcal{N}(0, \Lambda), \end{cases}$$

where we choose $\mu(r) \doteq \mathbb{E}\left[\int_r^0 f^2(z)\, \mathrm{d}z\right]$ in order to ensure that the prior mean at the origin is zero (*i.e.* $\mathbb{E}\left[\nu_r(0)\right] = 0$), while $r < 0$ controls the prior variance at the origin. In other words, we simply integrate $f^2$ from outside of the domain of interest.

Given the result (10), for stationary kernels this is equivalent to defining

$$\hat{\nu}^{(r)}(x) = \boldsymbol{w}^\top \boldsymbol{\psi}(x + r)\boldsymbol{w} - r \times k(0, 0),$$

with $\boldsymbol{w}$ and $k$ of Definition 2 and (7), respectively, and $\boldsymbol{\psi}(x)$ defined by (9).

This parsimonious approach allows to dispense with the *a priori* Gaussian intercept $\nu_0$, but is appropriate only for data which is known to lie on *e.g.* the positive real line, since $\nu_r(r) = -\mu(r)$ with probability one.

## C   Proof of Theorems 2 and 4

▷ Proof of Theorem 2 – (i) we rewrite (3) using $\chi = \nu^{-1} \circ g$ for a function $g$ that we want to elicit as the canonical link of a proper loss (*i.e.* negative the derivative of its Bayes risk). From (2), $g$ must satisfy

$$g^{-1}(x) = \frac{G'(-\nu^{-1}(x))}{G'(-\nu^{-1}(x)) + G'(\nu^{-1}(x))}, \tag{16}$$

Since $G$ is decreasing and convex, its derivative is negative increasing. $\nu^{-1}$ is increasing because $\nu$ is, and so the right-hand side of (16) is increasing with values in the $[0, 1]$ interval, furthermore having $g^{-1}(0) = 1/2$. Its inverse $g$ is therefore also increasing in the interval $[0, 1]$ and its derivative can therefore be used as weight function to craft a proper loss $\ell$ following *e.g.* [RW10, Theorem 1], yielding for this loss $g = -\underline{L}'$, as claimed for point (i). The proof of (ii) is immediate: as $G$ is convex and classification calibrated, it satisfies $G'(0) < 0$ [BJM06, Thm. 6], implying $G'_\nu(0) = \rho(0)G'(0) < 0$ since $\rho(x) \doteq 1/\nu'(\nu^{-1}(x)) > 0$, assuming wlog $|f| \ll \infty$ almost everywhere. This implies $G_\nu$ classification calibrated [BJM06, Thm. 6].

▷ Proof of Theorem 4 – Our proof uses (1) and (3). We consider a proper canonical loss $\ell(y, x)$ where $x$ denotes a real-valued prediction. Because a Bregman divergence is always convex in its left parameter, we have

$$
\begin{aligned}
\mathbb{E}_{(x,y)}\left[\ell(y, (-\underline{L}')^{-1}(x))\right] &= \mathbb{E}_{(x,y)}\left[D_{(-\underline{L})^\star}\left(x\| -\underline{L}'(y^*)\right)\right] \\
&= \mathbb{E}_{(x,y)}\left[D_{(-\underline{L})^\star}\left(\mathbb{E}_\nu\left[\nu(x)\right]\| -\underline{L}'(y^*)\right)\right] \\
&\leq \mathbb{E}_{(x,y),\nu}\left[D_{(-\underline{L})^\star}\left(\nu(x)\| -\underline{L}'(y^*)\right)\right] = \mathbb{E}_{(x,y),\nu}\left[\ell_\nu(y, \chi^{-1}(x))\right],
\end{aligned}
$$

as claimed.

## D   Inference with Integrated Squared Gaussian Processes (Additional Details)

Recall that we denote the parameters $\Gamma$ and the hyper-parameters $\Theta$. The Laplace approximation to the marginal likelihood is then

$$\log p(\mathcal{D}|\boldsymbol{\theta}) \approx \log q(\mathcal{D}|\boldsymbol{\theta}) \doteq \log p(\mathcal{D}, \widehat{\Gamma}) - \frac{1}{2}\log \det H.$$

in terms of the Hessian of (11). Optimising this expression with respect to $\boldsymbol{\theta}$ is non-trivial since $\widehat{\Gamma}$ depends on $\boldsymbol{\theta}$. We employ the generic approach from [SF06], which uses automatic differentiation to achieve the same result as — while avoiding the manual gradient calculations of — the usual approach in the GP literature (see *e.g.* [RW05, §3.4]).

We first use the total derivative to decompose

$$\nabla_{\boldsymbol{\theta}} \log q(\mathcal{D}|\boldsymbol{\theta}) = \frac{\partial}{\partial \boldsymbol{\theta}}\log q(\mathcal{D}|\boldsymbol{\theta}) + \left(\frac{\partial}{\partial \boldsymbol{\theta}}\widehat{\Gamma}\right)^\top \left.\frac{\partial}{\partial \Gamma}\right|_{\Gamma = \widehat{\Gamma}} \log q(\mathcal{D}|\boldsymbol{\theta}),$$

where we may show using the implicit function theorem that

$$\frac{\partial}{\partial \boldsymbol{\theta}}\widehat{\Gamma} = -H^{-1}\left.\frac{\partial^2}{\partial \Gamma \partial \boldsymbol{\theta}^\top}\right|_{\Gamma = \widehat{\Gamma}} \log p(\mathcal{D}, \Gamma).$$

In line with [SF06], we employ automatic differentiation to compute both $H$ (as a function of $\Gamma$), and then to differentiate $\log q(\mathcal{D}|\boldsymbol{\theta})$ (which is a function of $\det H$). In summary, the entire marginal likelihood maximisation procedure requires only *i)* the (trivial) implementation of the log prior and log likelihood functions, *ii)* automatic differentiation software such as [PGM+19] (to be invoked in three different ways[3]), and *iii)* , non-linear optimisation software.

|  |  |
|---|---|
| (a) GP Prior | (b) ISGP Prior |

Figure 3: Regression with a prior that is *a)* a Gaussian process and *b)* our integrated squared Gaussian process of section 3. We use (Laplace approximate) maximum marginal likelihood hyper-parameters along with our novel trigonometric kernel of subsection 4.3 in both cases.

▷ **Computational Complexity.** For likelihood functions of the form given at the top of Section 4.1 we may use (8) to compute $\nu(x_n)$. As a result, although inference under the ISGP prior may appear challenging due to the integral in Definition 2, the log posterior can be evaluated in only $\mathcal{O}(M^2N)$ time, where $M$ is the size of the basis and $N$ is the number of data points. This is the usual cost for sparse GP approximations with $M$ basis functions (or inducing points). The choice of *squared* transformation in Definition 2 makes this possible.

### D.1 Likelihood Functions

For concreteness, and to specify the parameterisations we employ, we complete this section by introducing the two univariate likelihood functions used throughout the paper.

▷ **Gaussian Likelihood for Regression.** Here we have $y_n \in \mathbb{R}$, $n = 1, 2, \ldots, N$, and

$$\log p(y_n|\nu(x_n), \Theta) = \log \mathcal{N}(y_n|\nu(x_n), \alpha^{-1}) \tag{17}$$

$$= \frac{1}{2}\log(\alpha) - \frac{1}{2}\log(2\pi) - \frac{\alpha}{2}\left(\nu_0 + \boldsymbol{w}^\top \boldsymbol{\psi}(x_n)\boldsymbol{w} - y_n\right)^2, \tag{18}$$

where for simplicity (and since our model is discriminative) we neglect to notate conditioning on $x$ both above and, as appropriate, throughout. This model is illustrated on the r.h.s. of Figure 3.

▷ **Sigmoid-Bernoulli Likelihood for Classification.** Here we let $y_n \in \{0, 1\}$, and

$$p(y_n|\nu(x_n), \Theta) = \text{Bernoulli}\left(y_n|(-\underline{L}')^{-1} \circ \nu(x_n)\right), \tag{19}$$

where $\text{Bernoulli}(y|\rho) = \rho^y(1-\rho)^{1-y}$, and we are composing with the logistic sigmoid $(-\underline{L}')^{-1}(\nu) = 1/(1 + \exp(-\nu))$. The above likelihood function may be expanded analogously to (18), to obtain a readily implementable form.

## E  Trigonometric Kernel (Additional Details)

▷ **Closed form** $k(x, z)$**.** Although we do not require it, the kernel is available in closed form.

Letting $d = c|x - z|$ we have

$$k(x, z) = \frac{b\left(ae^{\frac{i\pi d(2M+3)}{2}} + ae^{\frac{i\pi d}{2}} - e^{i\pi d(M+1)} - e^{i\pi d} - 2a^M e^{\frac{i\pi d(M+2)}{2}}\left(a\cos\left(\frac{\pi d}{2}\right) - 1\right)\right)}{2a^M e^{\frac{i\pi d(M+1)}{2}}\left(ae^{i\pi d} - (a^2 + 1)e^{\frac{i\pi d}{2}} + a\right)},$$

and for $M \to \infty$,

$$k(x, z) = \frac{b}{2}\left(\frac{a}{a - \exp(\frac{i\pi d}{2})} + \frac{1}{a\exp(a - \frac{i\pi d}{2}) - 1}\right).$$

▷ **Closed form $\psi$.** The integrals needed for $\psi(x) \in \mathbb{R}^{M \times M}$ of (9) are, for the pairs of sine terms,

$$\int_0^x \sin\left(\pi mcz\right) \sin\left(\pi ncz\right) \mathrm{d}z = \begin{cases} \frac{x}{2} - \frac{\sin(2\pi cmx)}{4\pi cm} & m = n \\ \frac{n\sin(\pi cmx)\cos(\pi cnx) - m\cos(\pi cmx)\sin(\pi cnx)}{\pi cm^2 - \pi cn^2} & m \neq n, \end{cases}$$

for the pairs of cosine terms,

$$\int_0^x \cos\left(\pi mcz\right) \cos\left(\pi ncz\right) \mathrm{d}z = \begin{cases} \frac{x}{2} + \frac{\sin(2\pi cmx)}{4\pi cm} & m = n \\ \frac{m\sin(\pi cmx)\cos(\pi cnx) - n\cos(\pi cmx)\sin(\pi cnx)}{\pi cm^2 - \pi cn^2} & m \neq n, \end{cases}$$

and for the mixed terms,

$$\int_0^x \sin\left(\pi mcz\right) \cos\left(\pi ncz\right) \mathrm{d}z = \begin{cases} \frac{\sin^2(\pi cmx)}{2\pi cm} & m = n \\ \frac{-n\sin(\pi cmx)\sin(\pi cnx) - m\cos(\pi cmx)\cos(\pi cnx) + m}{\pi cm^2 - \pi cn^2} & m \neq n. \end{cases}$$

# F  Nyström Approximation

Our *trigonometric kernel* of Section 4.3 is ideally suited to inference under the ISGP model, in that it admits efficient computation of the matrix $\psi(x)$ of (9). There is another more subtle condition which must be satisfied, however, in order for our Laplace approximate hyper-parameter optimisation procedure to be efficient. That is, only the spectrum $\Lambda$ may depend on the hyper-parameters of the kernel, while the basis $\varphi(x)$ must be fixed. Due to these requirements the tempting and popular Nyström approximation [Nys28] is generally insufficient, and our trigonometric kernel is therefore essential for tractable inference under the ISGP model.

In the remainder of this section we investigate an alternative approach based on the Nyström approximation to the kernel. In many GP inference problems, this approximation method is applicable to a rather wide range of kernels. Here however, we require the integral for $\psi(x)$, which is not generally available. Fortunately, as we now demonstrate, these terms are available in closed form for the Gaussian kernel.

Nonetheless, a drawback of the Nyström method remains as — unlike our Trigonometric kernel — the hyper-parameters of the kernel affect the basis $\varphi(x)$, not just the spectrum. This dependence renders the optimisation of our marginal likelihood approximation in Section 4.1 prohibitively expensive in general — for fixed hyper-parameters the Nyström method may be useful, however, and for completeness we derive the key expressions here.

The above drawback may be partly alleviated in certain cases, however. Indeed, given the univariateness of the ISGP, one length scale and one output scale parameter should suffice as the kernel hyper-parameters. Then, a finite difference approximation of the marginal likelihood derivatives should be within an (albeit very large) constant factor of the corresponding computation time for the trigonometric kernel.

▷ **General Setup.** The Nyström idea is to note that the $\varphi_i, \lambda_i$ pairs are eigenfunctions of the integral operator (see [RW05] section 4.3) on the reproducing kernel Hilbert space $\mathcal{H}(k)$ induced by $k$,

$$T_k : \mathcal{H}(k) \to \mathcal{H}(k)$$
$$f \mapsto T_k f \doteq \int_{x \in \Omega} k(x, \cdot) f(x) p(x) \, \mathrm{d}x,$$

where $p$ may be freely chosen provided it has an appropriate support. The idea of the Nyström approximation [Nys28] to $T_k$ is to draw $M$ samples $X = \{x_1, x_2, \ldots, x_M\}$ from $p$ and define the Monte Carlo approximation $T_k^{(X)} g \doteq \frac{1}{M} \sum_{x \in X} k(x, \cdot) g(x)$. Then the eigenfunctions and eigenvectors of $T_k$ may be approximated via the eigenvectors $e_i^{(\mathrm{mat})}$ and eigenvalues $\lambda_i^{(\mathrm{mat})}$ of $k(X, X)$, as (we abuse the notation so that $k(X, X)$ is an $M \times M$ matrix of kernel evaluations, *etc.*)

$$\varphi_i^{(X)}(z) \doteq \sqrt{M} / \lambda_i^{(\mathrm{mat})} k(X, z)^\top e_i^{(\mathrm{mat})}$$
$$\lambda_i^{(X)} \doteq \lambda_i^{(\mathrm{mat})} / M.$$

For our setting, we further require (in addition to the usual matrix eigendecomposition algorithm), the following integral

$$
\begin{aligned}
(\boldsymbol{\psi}(x))_{ij} &= \int_0^x \varphi_i^{(X)}(z)\varphi_j^{(X)}(z)\,\mathrm{d}z \\
&= \frac{M}{\lambda_i^{(\mathrm{mat})}\lambda_j^{(\mathrm{mat})}}\, \boldsymbol{e}_i^{(\mathrm{mat})\top} \underbrace{\int_0^x K(z,X)^\top K(z,X)\,\mathrm{d}z}_{\doteq \boldsymbol{\psi}_{k,X}(x)\in\mathbb{R}^{M\times M}}\, \boldsymbol{e}_j^{(\mathrm{mat})}.
\end{aligned}
\tag{20}
$$

▷ **Squared Exponential Kernel.** Although a Mercer decomposition is available in closed form (see *e.g.* [RW05]) for the popular kernel

$$
k(x,z) = b\exp\left(-\frac{a}{2}(x-z)^2\right),
$$

it turns out that the integrals we require for $\boldsymbol{\psi}$ are challenging for that decomposition. Fortunately the Nyström approximation is convenient, since the key term in (20) is given by

$$
\begin{aligned}
(\boldsymbol{\psi}_{k,X}(t))_{ij} &= \int_0^t k(x_i,z)k(x_j,z)\,\mathrm{d}z \\
&= \frac{b^2\sqrt{\pi}}{2\sqrt{a}}\exp\left(\frac{a}{4}(x_i-x_j)^2\right)\left(\mathrm{erfi}\left(\frac{\sqrt{a}}{2}(x_i+x_j)\right) - \mathrm{erfi}\left(\frac{\sqrt{a}}{2}(x_i+x_j-2t)\right)\right).
\end{aligned}
$$

(a) elements of $\boldsymbol{\psi}(\cdot)$ (for $M = 16$)

(b) kernel (for $M = 16$)

(c) kernel (for $M = 32$)

(d) kernel (for $M = 64$)

Figure 4: Visualisation of our trigonometric kernel of Section 4.3, and the $\boldsymbol{\psi}(z)$ of (9) induced by it. $M$ is (half) the number of basis functions. The remaining hyper-parameters are $a = 1.2$, $b = 1$ and $c = 1$. See the labels on the figures for details.

Figure 5: A toy regression problem with ISGP prior and Gaussian likelihood function. *Upper plot:* The posterior predictive distribution for $\nu$, our inferred monotonic function, along with the ground truth function and training data points. *Middle plot:* The posterior predictive distribution for our $f$ of Definition 2, which is the square root of the derivative of $\nu$, along with $\pm$ the square root of the derivative of the ground truth function. *Lower plot:* Similar to the middle plot but with the squared transformation included. We use maximum marginal likelihood parameters with the trigonometric kernel of subsection 4.3, and a kernel scale parameter of $c = 1/100$, so that the inferred functions are periodic with period 200.

Figure 6: A zoomed out version of Figure 5.

Figure 7: Visualising the log marginal likelihood for the problem of Figure 5 (and Figure 6). We computed the maximum marginal likelihood parameters using the method of subsection 4.1. Then we varied each hyper-parameter about this optimal value (represented by the greed dots), holding the others fixed. With the exception of the extremely flat lowest plot (for the prior variance $\gamma$ of the intercept $\nu_0$, which is expected to have a flat marginal posterior), the marginal likelihood optimisation finds a stable local minimum. Note that for clarity the vertical axis labels neglect to notate conditioning on certain variables.

(a) posterior latent source function $\nu(\cdot)$

(b) composition of the source function from (a) with the sigmoid $\sigma(y) = 1/(1 + \exp(-y))$

Figure 8: A toy classification problem with ISGP prior and Bernoulli likelihood function. *Upper plot:* The posterior predictive distribution for $\nu$ (our inferred monotonic function) along with the ground truth function and training data points (with binary labels represented by two distinct $y$-values). *Lower plot:* The posterior predictive distribution for the inverse link function $\sigma \circ \nu$, where $\sigma(\nu) = 1/(1 + \exp(-\nu))$ is a shorthand for the sigmoid function (inverse canonical link of the log loss).

(a) logistic regression

(b) GP linkgistic regression

(c) ISGP linkgistic regression

Figure 9: The inverse link function for logistic regression (upper), GP linkgistic regression (lower) and ISGP linkgistic regression (lower), on the *Fashion-MNIST* task of class 3 (*dress*) *vs.* the rest. The $x$-axis is the input to the (inverse) link function, which is equal to the output of the generalised linear model, *i.e.* $x_n = \boldsymbol{\beta}^\top \boldsymbol{z}_n$ as per Section 4.2.

## Footnotes

[3]These include: 1) differentiating $\log p(\Gamma|\mathcal{D})$ with respect to $\Gamma$ for *maximum a posteriori* optimisation, 2) computing the Hessian w.r.t. $\Gamma$, $H$, and 3) differentiating $\frac{1}{2}\log \det H$ w.r.t. $\Theta$ for *maximum marginal likelihood* optimisation — see [SF06] for the details.