[Reviews · NeurIPS 2020]

Review 1

Summary and Contributions: The paper proposes a methodology to jointly fit the parameters and the loss function of generalised linear models (GLM). Related approaches parameterise the loss in terms of an invertible link function (ie a monotonic function that maps [0,1] to R). While existing methodologies attempt to learn the link function directly, the authors propose to fit what they call the "source function", which is defined to be a differentiable strictly increasing function. It is shown in the paper that is it possible to construct a proper loss by means of the composition of a learnable source function and a canonical link. The source function is claimed to be more convenient from a computational point of view, as it only imposes monotonicity; the source can map R to R (as opposed to the link function). A GP regression scheme is proposed in order to perform inference for the source function. A particular prior is considered in the space of functions, which is the integral (up to x) of a squared GP (ISGP). The authors discuss the theoretical properties of the prior induced by ISGP, and they develop a Laplace approximation scheme to approximate the non-Gaussian posterior; the GP hyperparameters are treated by means of marginal likelihood maximization. They apply the scheme in temrs of a GLM, where the model parameters and the source function are trained using Expectation Maximisation (EM). In the experiments, the authors demonstrate the fitting of the source function and the actual learning of a classification loss on both toy and real world datasets.

Strengths: The paper discusses a novel approach to learn a loss function that I think would be of interest to the community. The parametrisation of the loss by means of the source function is novel and well-motivated. The authors use the results of recent literature to show that it is possible to reconstruct any proper loss function as composition of a canonical link and a source (Eq.4). Also, they propose a principled inference method to capture these source functions. There is an extended theoretical discussion over the properties of the proposed ISGP prior. It is shown that the sample paths of the prior are valid source functions, and it easily follows that the posterior samples are valid sources too. (Lemma 1, Theorem 3) In my opinion, this modelling of loss in combination with the inference scheme differentiate the paper from the previous literature. The parametrisation proposed is more 'tailored' to the requirements associated with fitting a loss function, and the experimental results seem to support this claim.

Weaknesses: My biggest concern regarding this work is readability. The authors could benefit from some extra space to provide some intuition regarding the concepts they introduce. I am a bit confused about Theorem 8. What is the distribution of \vu in the expectation on the left-hand side? Is this the true posterior distribution? If that is the case, then the loss induced by the approximate posterior is an upper bound to the loss of the true posterior, which would be remarkable.

Correctness: I think that the claims and the mathematical derivations in the paper are correct. The methodologies used to perform inference, including the Laplace approximation scheme and the expectation maximization algorithm are appropriate for the problem at hand.

Clarity: I think the paper is difficult to understand for machine learning researchers that are not very familiar with the topic. Also, I appreciate the popular culture reference of the title, but I don't think it clearly reflects the content of the paper.

Relation to Prior Work: The authors clarify that the paper is effectively an extension of a very recent piece of literature [NM20]. Compared to the work in [NM20], the authors of the current paper employ a Bayesian regression method (ie the ISGP scheme) in the place of the optimisation scheme of [NM20]. Also, inference is performed on the source function, rather than the link as in previous works. These contributions are clearly stated.

Reproducibility: Yes

Additional Feedback: Post Rebuttal Feedback: I thank the authors for their response. Personally, I am in favour of a more technical title. Regarding to Vapnik's quote, Yann LeCun refers to it as an "inside-inside" joke. So I think that the current title is way too cryptic.


Review 2

Summary and Contributions: This paper proposes a method to augment the proper loss functions when evaluating a classifier. Instead of using single proper loss on the fitted function, this paper considers using a source function to map the fitted function to a set of latent functions before calculating the loss, thus allowing the final loss to be augmented. A monotonic GP is applied to model the source function to ensure the invertibility between the fitted function and the latent functions, thus keeps the properness of the original loss.

Strengths: 1. The direction of the work is interesting. It has been a default setting in the machine learning community to pick a single proper loss for probabilistic classifiers. This paper points out a way to enrich a given proper loss without affecting its properness. 2. This proposed method is constructed and proved on a set of related work on proper losses/margin losses/link functions, hence is theoretically sounding.

Weaknesses: 1. While both the research direction and solution are interesting, the title and ceratin statements in this paper seem to be over-claimed. First of all, the title is "All your loss are belong to Bayes", which is not very informative on the paper's content. To my understanding, given a particular proper loss, the proposed method allows augmenting the given loss via inferring a set of source functions. This means the final obtained loss is still depending on the original proper loss. So the proposed method is unlikely to perform inference on different types of proper losses, such as Brier score and log-loss (I am not saying impossible, but it requires further mathematical demonstration to see on what circumstances different proper losses can meet). Also, the proposed method seems to only work on a binary setting, which is limited compared to the general setting of proper scoring rules. 2. Although this paper is more on the mathematical and theoretical side, the execution on evaluation is relatively weak. All the classification experiments only use the logistic link, which is only limited to a particular kind of proper loss. Experiments on more related loss/link functions would make the results stronger as the paper claims to contribute a general framework for inferring proper losses. Also, while the paper suggests their approach improves "consistency, robustness, calibration and rates.", none of these benefits are formally defined and checked in the experiments. For instance, outputting calibrated probability has been a key property of classifiers and is closely related to proper losses. It would be better if the author can indeed to check ceratin metrics like expectation calibration error to evaluate the proposed approach.

Correctness: As described above, I agree that the proposed method is useful and has potential benefits from existing approaches. However, the paper title and some statements in the paper might need adjustments.

Clarity: While a reader with a related background can understand most of the paper, this paper is quite heavy on math, and some notations can be confusing through the reading. It might be useful to add a table of related notations in the supplementary.

Relation to Prior Work: Yes, the discussion on existing work and the differences from them are relatively clear.

Reproducibility: Yes

Additional Feedback: I appreciate the efforts on the theoretical work in this paper; the general ML community might benefit more if this paper can include more empirical analysis /demonstration on the difference between the proposed approach and existing proper scoring rules. At the moment, it seems to be a complex and expensive method without detailed pros and cons. It would also be better if the authors can be more accurate on the paper title as the current one seems too broad. ===========AFTER AUTHOR FEEDBACK===================== Following the discussion with other reviewers, I increased my score to 6 as this paper provides a nice extension within a particular area (e.g. NM20) While I do appreciate the idea and agree it brings interesting insights to investigate proper losses, I am still not fully convinced if I should switch to the proposed approach from existing proper losses. (e.g. providing higher accuracies might also make the model vulnerable to predicting uncalibrated probabilities). I also feel there is some miscommunication between the authors and me, as shown in the feedback. Please see some further explanations below. 1. The authors replied that their approach could work with any proper loss. However, I wasn't questioning this. I was wondering if the proposed method can build links between different proper losses. For instance, Brier Score and Log-loss are quite different losses, and I thought maybe the proposed augmentation could somehow merge them to give a more generic view on these losses. 2. The authors said the new loss could work on a multi-class problem via one-vs-rest / 1-vs-1, which agrees with my comment that the proposed loss can only work on a binary setting. Proper losses like Brier Score supposed to be true multi-class and doesn't require such repeated training to work on more than two classes. 3. The authors suggested their method is not expensive as the inference can be quick. My point was the proposed method is more expensive than simply applying existing proper losses, which doesn't require any optimisation steps at all.


Review 3

Summary and Contributions: This paper proposes a novel approach to estimate the loss function in classification problems, and it derives theoretical conditions and analyses about the problem of loss estimation.

Strengths: This is an important problem, for which some approaches have appeared in the earlier literature of statistics. This paper proposes a fresh look at the problem and some solid theory around the proposed approach.

Weaknesses: I believe that the presentation of the paper could be improved. I also think that the paper could benefit from the use of more modern inference techniques for Gaussian processes. The experiments could also feature some more compelling examples where inference for the loss could lead to considerable improvements in performance. Finally, a clear algorithmic breakdown of the proposed approach would help clarifying what the Authors implemented exactly.

Correctness: As far as I can see, the method is correct. The empirical methodology could be improved by providing more comparisons and analyses.

Clarity: Title and abstract would benefit from a rewriting. I find the title nonsensical.

Relation to Prior Work: I think that the paper is well positioned within the literature on the topic.

Reproducibility: No

Additional Feedback: This paper challenges the common practice of choosing a loss function for supervised problems from the outset, and proposes an inference approach to the loss function. I generally like the idea proposed in the paper, and I would like to see this published. However, I have a number of comments for the Authors, which I hope will be clarified/discussed in the rebuttal and discussion phase: - The presentation of the paper could be improved. I think that the title could better reflect what the paper is about and the abstract could do with a bit of rewriting to be a bit more clear and specific about what the paper proposes. - From reading the development of the presentation, it seems that the proposed approach is proposed for classification problems only (but the experiments consider regression as well). If this is the case, this should be stated from the beginning. Otherwise, it should be said that the working example in the paper is classification, but that the approach can be generalized to any supervised learning task (in this case it would be great to showcase an example involving other types of data, e.g., counts). - I wonder whether the uncertainty estimation of the loss, given by the GP, is actually used anywhere. I guess it would be nice to analyze this; I speculate that in the case of small data sets it could give some interesting insights. - I think that the problem that the Authors are investigating is interesting. After reading the paper, however, I recalled a debate about an attempt by M Aitkin to infer the likelihood function and the debate that followed, championed by Gelman, Robert, Rousseau (https://arxiv.org/pdf/1012.2184.pdf). I haven't closely followed the developments of this debate, but I believe that this is definitely something that needs careful consideration if one approaches the related topic of loss inference. - Somewhat related and against the previous point, the approach of Snelson et al. NIPS 2004, where the output of a GP is warped using a function that is learned at training time, could also be something interesting to discuss/compare against. - The presentation of the integrated squared Gaussian process (ISGP) is a bit misleading. I guess the GP formulation is useful to directly derive the integration of the square of the function, but in practice this is just a linear model with a set of basis function. I wonder about the need to deal with such a construction and whether there are no easier ways to do so. For example, a linear model with monotonic basis function and positive weights would allow one to infer monotonic functions. The positivity on the weights could be handled with a log-transformation and inference could be carried out using modern variational inference. The ISGP seems to form an important part of the proposed method, and I'm just wondering whether it is strictly necessary and something easier couldn't achieve the same effect. - Related to the previous point, the introduction seems to imply that the proposed approach is nonparametric (although after re-reading the intro a few times, this is not explicitly stated, so it is just an impression given by the wording of a few sentences). Maybe it would be worth stating that the proposed approach in based on a parametric approximation of a GP. - The choice of basis functions seems to fall back on something that the GP community would recognize as random features (for the RBF kernel). The inference method used here seems a bit outdated compared to recent approaches to inference for GPs, such as variational inference. I wonder why this was not attempted or discussed. Also, maybe a few references on this, such as Lazaro-Gredilla et al., JMLR 2010 and Cutajar et al., ICML 2017, could improve the positioning of the paper wrt to the GP literature. - I find it unclear why Sec 4.1 is dedicate to the univariate case (of what?). I don't see why it is important to make a distinction with the multivariate case, which is not explicitly labelled or discussed in the sections. - The experiments do not seem to be so compelling in supporting the need to infer the loss function, so I'm worried that this paper might not make enough impact in the community. One idea could be to consider the problem of learning a loss function for models beyond generalized linear models. After all, it should be possible to learn a model and the loss end-to-end with the help of automatic differentiation. ************ After the rebuttal Many thanks for your feedback, which clarifies a few points. I think it is ok to keep this title as long as it is explained in the main text - people that know about Vapnik's quote will realize the connection, whereas people that don't know about it will be able to make the connection from the main text. I hope that the reviews will serve as a way to strengthen the camera ready. We gave a number of suggestions which I think are valuable in improving clarity. In particular, I hope that the Authors will improve the part on ISGPs and follow our suggestions to position this in the best possible way within the literature on GPs. I still think that the experimental validation is a bit on the weak side - the results show that GLMs can be improved by learning the loss, but the reader is left wondering whether a more complex model with a fixed loss would achieve the same effect. I speculate that this is the case (see results on multiclass MNIST in Fig 2b which are barely above 90%). I appreciate the effort on the theory side though - I'm not an expert on this particular line of works, but given that this is something people have worked on, this paper adds some new insights, and in this respect I have no reasons to argue against acceptance.


Review 4

Summary and Contributions: The paper learns the loss function in a Bayesian manner by decomposing the loss function into a link and a source, and then using the ISGP to model the source. Inference with ISGP is done via Laplace approximation, and a trignometric kernel is proposed as a practial manner to implement the model.

Strengths: The paper is has a strong foundation in theory (section 2). It proposes a GP based model to implement the theory, with a suggestion of the inference method. This will serve as a useful reference point for future work in this area. One novel contribution can be additionally stated in the introduction: the introduction of ISGP, which can be of independent interest.

Weaknesses: W1. Is the trigonometric kernel universal? and this is not clearly stated nor proved in the paper. W2. Althougth universal kernels are "nice to have" (Theorem 3), are they really crucial to the applicability of the model? --- the authors have not expressed their opinon on this explicity, though Theorem 2ii seems to imply that it does not matter for classification. W3. The paper only provides the trignometric kernels, and it seems that other kernels are rather hard to use. Does this limit the applicability and flexibility of the method? W4. The OU kernel has rather easy eigenfunctions and eigenvalues (see appendix B.6.2. of Cha10). Though it is not a universal kernel according to Theorem 17 of MXZ06, it should also be investigated empirically. W6. Since the paper focused on losses, the experiment on isotonic regression in section 5 seems like a distraction --- it can be moved to the supplementary material. I would prefer more experiments on learning the loses, and also analysis of the results. See also C5 below. Chai. Multi-task learning with Gaussian processes. PhD Thesis. 2010.

Correctness: Mostly. Though it is unclear if the trigonometric kernel is universal, and if not, then the theory (Theorem 3 requires universal kernel) is currently not supported by a model.

Clarity: Mostly. Some comments C1. Figure 1 rather hard to interpret because the terms are not explained within the caption itself. The general term "Correspondence" in the caption does not help. C2. In line 137, it is good to inform the reader where "later" is. Seems to be Theorem 5 and section 4.3, but I am not sure. C3. In the GP community translation invariant (line 172) more commony known as stationary. C4. Line 233: If the prior is GP, then this is not a construct for proper composite loss, as implied by Definition 1 (line 120) and noted on lines 291/291. It is best to bring upfront at line 233. The outcome is simply a GP classification model with a very perculiar kernel that first projects $x$ onto a line. C5. The references to the the supplementary material (SM) in Section 5 is makes the main paper totally dependent on reading the SM, which defeats the entire purpose of having a proper main paper and a SM. If the authors deem the material in SM so important, they so be moved to the main paper. In fact, Figures 8 and 9 should be moved to the main paper since they are central to the paper.

Relation to Prior Work: Suggest the authors to also discuss the following two work in relation to theirs: Edward Snelson, Carl Edward Rasmussen, and Zoubin Ghahramani. 2003. Warped Gaussian processes. In Proceedings of the 16th International Conference on Neural Information Processing Systems (NIPS’03). MIT Press, Cambridge, MA, USA, 337–344. Miguel Lázaro-Gredilla. 2012. Bayesian warped Gaussian processes. In Proceedings of the 25th International Conference on Neural Information Processing Systems - Volume 1 (NIPS’12). Curran Associates Inc., Red Hook, NY, USA, 1619–1627.

Reproducibility: Yes

Additional Feedback: A1. Not pertaining just to this paper, this area of work seems to be focused on the latent model being linear, but with a flexible loss function --- GLM models. How does it compare with a flexible latent model (e.g., GP) but a fixed loss function? Comments after rebuttal =================== Thank you for the reply. I have just one more comment to make. It seems to me that the Nystrom method is more general than the trigo kernel, which seems to be an invention just for ISGP. In this case, I would prefer the Nystrom to be in the main text and the trigo to be in the SM if there is a lack of space. Also, I've forgotten to mention that you should also relate the trigo kernel to some of the Fourier expansion approximations of GP in the literature.

[Author Response · NeurIPS 2020]

Thanks for many comments ! Below, reviewers shown by their CMT # (①–④) (excerpt of comments in *blue*), reference
to lines of submitted draft as L$xxx$, (OP) = Open Problem(s), (SM) = Supplementary Material, (CR) = Camera Ready.
▷ ①②③. Our title originates in Vapnik's quote: `https://tinyurl.com/y5jnurau`. Roughly, Vapnik's Structural
Risk Minimization (as in the pictured equation) embeds uncertainty due to both sampling and model choice – justifying
Vapnik's quote. **However** the loss in SRM is ad-hoc and as we write, statistical decision theory has long made an
*intensional* case for its choice. Our approach suggests that a Bayesian framework *could* be better than the frequentist
ones [NM20] to cope with this uncertainty – hence our title. We take the title feedback seriously and can elaborate in
the additional CR page OR opt for a technical title, even when the connection with Vapnik would wane in this change.

▷ ① *[...] benefit from extra space [...]*: Thanks, we plan to use the extra CR page as detailed in this rebuttal. *[...]*
*distribution of ν [...]*: This is the approximate posterior over source functions and is conditional on our model in the
usual Bayesian manner; the r.h.s. merely features the parameters of the posterior mean function – *c.f.* Lemma 7.

▷ ② *While both the research [...] proper losses can meet*: Our method does work with **any** proper loss in which
the source is embedded. Brier score, log-loss (*our experimental choice*). ②'s impossibility concern highlights a key
feature and is covered by our Theorem 3. We gratefully offer rebuttal-size argument restricted to ②'s examples, which
are symmetric proper (SP) with invertible links. 2 steps: (i) elicit composite link $\chi$ such that $(-\underline{L}_{\mathrm{us}})^\star(-y\chi(u)) =$
$(-\underline{L}_{②})^\star(-yu), \forall y \in \{-1, 1\}, u \in [0, 1]$. Invertible link implies $(-\underline{L})^\star$ strictly monotonic and thus invertible, and we
get $\chi(u) = -y \cdot ((-\underline{L}_{\mathrm{us}})^\star)^{-1} \circ (-\underline{L}_{②})^\star(-yu) = -((-\underline{L}_{\mathrm{us}})^\star)^{-1} \circ (-\underline{L}_{②})^\star(-u), \forall y \in \{-1, 1\}$. The last identity
holds because SP losses satisfy $(-\underline{L})^\star(-x) = (-\underline{L})^\star(x) - x$ [NN08, eq (10)]. (ii) we get the source $\nu$ from $\chi$ and
$\underline{L}_{\mathrm{us}}$ using eq. (3) in our paper, which can be expressed by an universal kernel. QED. *Also [...] only work on a binary*
*setting*: We **respectfully disagree**: it works without modification in multiclass multilabel case by using a 1-vs-all or
1-vs-1 multiple coding. *Although this paper [...] check ceratin metrics like expectation calibration [...] proposed*
*approach*: There is probably a **misunderstanding** here. Consistency, calibration, rates are **formal** properties of a loss.
For example, calibration is equivalent to a negative derivative in zero of the margin loss, which guarantees "label
consistency". It is not an experimental property. We are happy to make L96-L97 and ref. [BJM06] more explicit using
CR. Only robustness could be checked but it would fairly deserve a paper of its own; we are happy to push it as OP. *[...]*
*it seems to be a complex and expensive method [...]*: We **respectfully disagree**: *c.f.* Sec 5 of [NM20] – their *fastest*
option is $\mathcal{O}(N \log N)$ *per iter* but at the expense of a very involved data structure ($\mathcal{O}(N^2)$) without). Our simpler to
implement $\mathcal{O}(M^2 N)$ per EM iter (L529) admits control of $M$, a meaningful knob to drive complexity below [NM20].

▷ ③ *[...] a clear algorithmic breakdown [...]*: The paper being already dense in content, we are happy to push additional
pseudo-code in SM. *I wonder [...] interesting insights*: We integrate out $\nu$ for prediction similarly to a GP classifier.
*[...] M Aitkin [...] debate [...]*: Thanks ! See the motive behind our title above as a token that inference on the loss
+ model could save a few "Bayesian eggs" (we can discuss in CR). *I wonder about the need to deal with [the ISGP]*
*[...]*: We give extensive references to alternatives on L52-L55, all of which involve certain drawbacks compared to our
ISGP. However, we believe that the reviewer's suggestion is both novel and very interesting for this application, thanks !
*[...] worth stating [...] parametric approximation of a GP*: Agreed, thanks (although much GP inference exploits a
parametric approximation) . *I find it unclear why Sec 4.1[...]*: Agreed on the poor headings. Here univariate $\approx$ ISGP &
multivariate $\approx$ our complete model; we will improve for CR. *The experiments do not seem [...] One idea [...]*: We
respectfully disagree wrt the "learning the loss" problem: see Table 1 in [NM20] and justification in our L34-L35.
**However** we do agree that building up above GLMs is probably the best way to further widen the gap, *yet* one must
keep in mind the complexity cost, so it is more than about more complicated models (see ▷ ② above). *[...] inference*
*method [...] outdated* : Agreed, however efficient variational inference for the ISGP is an OP due to intractable integrals.

▷ ④ (ref tokens much appreciated) *A1*: This is an interesting OP, thanks; our flexiblility in the loss may ease the burden
of underlying model capacity. *W2*: The purpose of Th.3 is to show that kernels **do** fit to solve our problem. Considering
the complexity constraint for all approaches (see ▷ ② above), one might benefit to yield on universality for practical
purposes – and our experiments display that this is more than fine, in line with our intuition given the vast richness of
the space of losses we consider. *W1+W3*: The universality of the trigonometric kernel is an OP, **but** we do offer an
alternative Nyström based method in Appendix F which we show to be readily applicable to the (universal) Gaussian
kernel (satisfying the conditions of Theorem 3). Moreover, pending the integral of L583, it could be not just the
Gaussian kernel that would be covered. Plus, we may have slightly undersold the Nyström method from a complexity
standpoint (at least form a theoretical standpoint): given the univariate-ness of the ISGP, one length scale and one
output scale parameter should suffice as the kernel hyper-parameters. Then, even a finite difference approximation
of the marginal likelihood derivatives should be within an (albeit rather large) constant factor of the corresponding
computation time for the trigonometric kernel. *W4*: Agreed, great pointer ! **But** we caution that those eigen-functions
depend on the hyper-parameters, making inference more expensive as above. *C1, C2, C3* Agreed, thanks ! *C4*: Well
spotted for the GP, thanks ! **But** of course inapplicable to our ISGP. *C5*: See comments on experiments and W6 above –
we will use the additional page to alleviate dependencies on SM, and we shall add refs in Sec 6.

[Meta-Review · NeurIPS 2020]

After the discussion phase, there is a consensus among the knowledgeable reviewers that this is a good paper that warrants acceptance. The paper is therefore accepted as poster. However, there are a few points that I urge the authors to take into consideration when preparing the camera-ready version. In my opinion, the reviews, especially that of R3, are quite thorough already, so I hope the authors will consider some of the important suggested changes. Hence, I only emphasize the important points here. First, please improve the readability and presentation of the paper and make sure that it is accessible to the machine learning folks. Second, please re-consider the title of the paper. The title should clearly reflect the content of the paper. Please, avoid the title that is too cryptic (but it's also fine if the authors decide to keep the title as is).